# Supporting Students with Neurodevelopment Disorders in School Health Care—School Nurses’ Experiences

**DOI:** 10.3390/ijerph17165752

**Published:** 2020-08-09

**Authors:** Andrea Berglund Melendez, Maria Malmsten, Eva-Lena Einberg, Eva K. Clausson, Pernilla Garmy

**Affiliations:** 1Faculty of Health Sciences, Kristianstad University, 291 88 Kristianstad, Sweden; andrea.melendez8608@gmail.com (A.B.M.); mariahove@yahoo.se (M.M.); evalena.einberg@hkr.se (E.-L.E.); eva.clausson@hkr.se (E.K.C.); 2Clinical Health Promoting Centre, Lund University, 221 00 Lund, Sweden

**Keywords:** children, adolescents, school health care, school nurses, neurodevelopmental disorders, stress, pain

## Abstract

Students with neurodevelopmental disorders (NDDs) are present in every school, and most likely, there are a few students in every class. School health care is responsible for providing support to all students, especially those with special needs. The aim of the study was to describe school nurses’ experiences of supporting students with NDDs. A qualitative method consisting of seven focus group interviews (that included a total of 35 school nurses) in Southern Sweden was conducted. Three themes were identified in the findings: helping students with NDDs to interpret sensations, detecting early signs of distress among students with NDDs, and using an inclusive design for health education. This study highlights the importance of school nurses in identifying the needs of students with NDDs and promotes a person-centered approach to achieve a healthy and safe learning environment for all students.

## 1. Introduction

Neurodevelopmental disorders (NDDs) are rising among school-aged children and adolescents, especially in those with autism spectrum disorders (ASD) and/or attention deficit/hyperactivity disorder (ADD/ADHD) [1,2]. Coping with stress and pain can be challenging for all children and adolescents, especially those with NDDs [3]. Students with NDDs should have their social skills and emotion management skills identified as early as possible, because they are at higher risk of decreased psychological, social, and physical well-being [3]. School health care has a crucial role in supporting children and adolescents with NDDs [4,5,6], for example, to develop individual health and education plans for students with NDDs.

### 1.1. Background

NDDs are multifaceted disorders characterized by impairments in cognition, communication, behavior, and/or motor skills related to brain development. Intellectual disability, communication disorders, ASD, and ADD/ADHD all fall under the umbrella of NDDs [1]. Diagnoses of ADHD and ASD are becoming increasingly more commonplace [2], although a significant difference exists between these disorders. A diagnosis of ADHD implies hypo- or hyperactivity and impairments with attention and impulse control, while children with ASD may have social difficulties and communication problems. Children with NDDs can range from severely impaired to highly functional. Both children with ASD and ADHD speak in shorter sentences and are more likely to repeat themselves than neurotypical children [7]. Those with ADHD have an increased likelihood of engaging in risky behavior and consuming alcohol, tobacco, and other illicit substances in their adolescent years [8].

If the family has a good understanding of NDDs and effective coping techniques, as well as aid from school health care, they will be in a better position to help the children. Often families with children with NDDs feel that they lack knowledge of general healthcare, which can lead to a lack of support from healthcare providers [9]. Bellando and Lopez [4] have identified six steps that provide a framework for the process of developing an individual health plan for students with NDDs. The first step involves defining the nurse’s role in the presenting problem. The second step includes characterization of the present problem. In the third step, information about the medical aspects of the current problem has to be gathered. In the fourth step, information about the developmental issues (cognitive, language, motor delays, and/or sensory issues) must be gathered and analyzed; then, it must be determined how this information contributes to the present problem. In the fifth step, the psychological and behavioral aspects are explored. The sixth and final step involves implementation of the individual health plan. The plan should be written free of medical jargon and easy to understand. If possible, the plan should be reviewed by the school health care team. It is essential to establish a timeline to gather data, get back together to modify the plan, and discuss the next treatment steps. Guidelines should be established regarding what constitutes a medical emergency and when health professionals need to be contacted. In addition, the student and the legal guardians should be involved in the health plan [4].

The law mandates that every school in Sweden have a healthcare service whose main purpose is to promote health and prevent ill-health among its students [10]. School nurses in Sweden are responsible for an average of 400 to 600 students, and their main tasks involve regular health check-ups, health dialogues, immunizations, and health education [11].

Studies emphasize the importance of close collaboration between school nurses and teachers; lack of such collaboration can have deleterious effects on students with NDDs (e.g., teachers failing to recognize signs of distress) [4,5,6].

Supporting students with special needs, such as children with mental health problems [12] or refugees [13] has been found to be both meaningful and challenging for school nurses, although there seems to be potential for improving the work through professional development with mentorship and enhancement of new knowledge. Our hope is that this study will help to guide early career school health professionals to access tools and suggestions regarding how to support students with NDDs. We also hope that this study can be used as a discussion template among experienced school health professionals in a community of practice to give the best possible support to students in need.

### 1.2. Aim

The aim of the study was to describe the experiences of school nurses in supporting students with NDDs.

## 2. Method

### 2.1. Design

This qualitative study used focus group interviews with school nurses. It was approved by the Regional Ethical Review Board in Lund, Sweden (EPN 2018/842).

### 2.2. Settings

The study took place in six municipalities in Southern Sweden. Seven focus groups were held with school nurses from schools in both urban and rural areas with a variety of socio-economic backgrounds among the student population in the catchment area. Each school nurse was responsible for approximately 400–500 students, and all nurses worked in public schools with students aged 6–15 years. A semi-structured interview guide was used. The questions were as follows, “Please describe how you recognize stress and pain among students with NDD” and “Please describe how you provide support to students with NDD”. Follow-up questions were asked to obtain a more in-depth description (e.g., Could you tell me more? What do you mean? How is this done?). The focus group interviews were conducted from May to October 2019, and each focus group interview was 70–90 min in duration. School nurses chose the time and place for the interviews, and all interviews were conducted during the day at the school nurse′s workplace. All focus group interviews were conducted in adequate rooms that were free of noise and external interruptions. The interviews were audio-recorded and transcribed verbatim by the first two authors. The intention was to have one moderator and one observer in each focus group interview. This was the case in five focus group interviews; however, in the two last sets of focus groups, there was one moderator, but no observer. The authors A.B.M., M.M., P.G., and E.-L.E. took turns moderating and observing. All authors are registered nurses with extensive knowledge of NDDs. P.G., E.K.C., and E.-L.E. have a PhD and several years of experience as school nurses. A.B.M. and M.M. were Master students at the time of data collection.

### 2.3. Study Population

The process of recruiting participants for the focus groups began with sending written information to all school nurses (*n* = 100) at public schools in the included municipalities about the aim of the study, the course of action, and ethical considerations. We did not ask why school nurses did not want to participate; however, some school nurses provided reasons for not participating. The main reason was lack of time, and a few school nurses claimed that they had limited experience of working with children with NDDs. Those who consented to participate contacted the researchers to arrange a date and location to hold the focus group interview. Each focus group consisted of school nurses from the same municipality, and the participants were, therefore, acquainted with each other. In the last sets of focus group interviews, no new information was obtained, and, therefore, the data collection was considered sufficient after 7 focus groups. The focus group sessions included three to seven participants. A focus group interview takes advantage of group dynamics to access richer and more nuanced information. The life experiences of each participant inform the other participants’ perceptions and understanding of the phenomena in question [14]. All school nurses had a relevant specialist nurse education at a Master’s level, as district nurses (*n* = 19), pediatric nurses (*n* = 13), or school nurses (*n* = 3). Work experience as a school nurse ranged from 1.5 to 36 years. All school nurses had extensive experience of working with students with NDDs.

### 2.4. Analysis

A qualitative content analysis inspired by Graneheim and Lundman [15] was undertaken. An important characteristic of qualitative content analysis is that it focuses to a large extent on the subject and context, emphasizing differences and similarities within the text. The deriving of the meaning unites followed a standardized procedure [16]. The first two authors began reading through the text several times, first independently and then together to obtain a fuller sense of the text. They then identified meaning units separately and compared their results to see whether their conclusions aligned and to ensure that no important data were overlooked. The meaning units were then condensed, abstracted, and labeled with a code. The results were then sorted into three themes by all authors. The analysis was discussed among all five authors until consensus was obtained.

## 3. Results

The school nurses emphasized the importance of taking the students’ feelings and complaints seriously. Many of the students with NDDs complained of pain, frustration, and aches that they had difficulty describing. The pain may be physical, but it could also be psychosomatic or entirely psychological in nature. The school nurses aimed to convey a caring, receptive atmosphere where students can be heard and understood. By being calm and confident, the school nurses aim to reassure the students that they are in a safe environment to voice their concerns and receive appropriate support. Three themes were identified in the results: helping students with NDDs to interpret sensations, detecting early signs of distress among students with NDDs, and using an inclusive design for health education.

### 3.1. Helping Students with NDDs to Interpret Their Sensations

The school nurses reflected that stress and pain in children with NDDs could be expressed in frustration and irritation, and that the school nurses had an important function in helping students to interpret their sensations. Stress and pain could easily lead to an outburst or self-harm. The frustration is often that students with NDDs have difficulty expressing what they feel and what they need:
The outbreaks often occur because children with NDD feel stressed but cannot express that they are stressed. They then act in a way that others do not think is okay. It is a frustration not being able to express what it is that they feel. It’s as if there is a feeling in my body, but I cannot express what that feeling is. It just bubbles up as something outward. (Focus group No. 1)

Many students with NDDs have a more intense feeling experience, which makes them feel pain more intensely and faster. The ability to determine how badly it really does hurt is delayed, so students with NDDs feel very strongly that they cannot do anything about their pain. The school nurses then experienced that they played an important function in helping the student to maintain in the situation, asking, “How is the pain?”, “How does it feel?”, “How dangerous is it?”, and “Describe the pain.” The nurses emphasized the importance of taking the student′s pain seriously, allowing for a rather slow course. They emphasized the importance of using sensory examination, to feel and squeeze a little here and a little there, taking it easy and using some distractions. It usually does not take long for students with NDDs to pull themselves together, and then the school nurse together with the student can evaluate and see how they should proceed when they are on the same level.

The school nurses also indicated that it is important for students with NDDs not only to talk, but also to use more senses when the student raised a concern, for example with pain in the arm. The school nurse may need to look and feel carefully at the place, to help the student understand what is happening. The school nurses also emphasized the importance of helping students with an NDD to understand that all pain or physical and emotional sensations need not be dangerous, but that it is part of being human to feel different emotions and sensations. One example that often came up was muscle fatigue. It can be difficult to understand that an activity that is good, i.e., participating in a physical education (PE) lesson, can cause pain the next day.

The school nurses experienced the importance of how to help students with NDDs to express themselves and that misunderstandings are easily created with children with NDDs. A school nurse told of a child with an NDD who had toppled over, and the PE teacher said to the child, “Oh, now you have a hole in your head.” The child was in complete despair thinking there was a hole going straight into the brain. Another child had been told that his broken arm was “off” and then thought that his hand was gone. The school nurses said that students with NDDs often have different pain control compared to other children, which means that they can be extremely sensitive to different sensations, for example, if the food is too hot, or how their clothes feel on their body. The pain can then become gigantic, almost overwhelming. If a student with an NDD has an arm pain, the pain can be felt throughout the body and make the whole body unable to move:
If I tell a student with NDD that he can go even if he has an arm injury, the student responds, “No it is not possible.” They cling to the pain, and do not think that although it hurts the arm, the legs work. That connection is missing in many children with NDD. (Focus group No. 2)

The school nurses also expressed that an important task was to help students with NDDs to put their pain in a reasonable context. A student with an NDD, who was a goalkeeper when they played football in PE, went in to see the school nurse and say that he had been beaten. However, as it turned out, he was hit by a ball in the game.

### 3.2. Detect Early Signs of Distress among Students with NDDs

The school nurses mentioned that it was crucial to detect early signs of distress among students with NDDs, because it otherwise could lead to problems for both the child and the peers. Many children with NDDs are afraid of pain, and this fear causes some barriers to disappear, and the pain may often be displayed as anger.

The school nurses experienced that it is a challenge to pay attention in time when children with NDDs feel stressed. Often distress is not noticed until the issue has gone too far:
…and the tendency of the introverts to be missed is much greater than the extroverts. Because it is not visible when they are closing down, it is not noticeable if at the same time someone else is getting an outbreak. (Focus group No 3)

The school nurses expressed that they may have missed that there was a previous change and then explained the child′s behavior with the diagnosis and the fact that it was a bad day. However, they may have missed that it was a bad day today, yesterday, and the day before yesterday. It takes a little time to see the connections, and see those early signals that are easy to miss. The school nurses believe that it can be things that we normally do not think of as stressful, such as something changing or coming to the fore, factors like the sun shining brightly. It may be other things; it does not have to be something obvious like having three tests. The school nurses say that students with NDDs in stress need more support than usual based on their difficulties. Under stress, they forget more easily and act faster on emotions and impulses and changes in emotions. Other early signs of stress in students with NDDs are extreme fatigue. The fatigue can cause children with NDDs to stay home from school.

The school nurses mentioned that early signs of distress could be students’ spontaneous visits to the school health care with symptoms of pain. The school nurses express that they often find that students with NDDs who come in with pain often have an underlying reason that is the real concern. The school nurses feel as if the students may be testing them before telling them the real reason for the visit.

The school nurses highlighted that it was important to pay attention to early signs of distress that could lead to substance use among students with NDDs. The school nurses felt that students with NDDs used fewer painkillers than other children, but the risk was higher for these children to try tobacco, alcohol, or other drugs. They thought that because many children with NDDs were impulsive, they were more likely to try these substances.

The school nurses experienced difficulties in detecting early signs of distress among girls with NDDs. The school nurses thought that since many girls with NDDs did not interfere in the classroom and may not even visit the school nurse to the same extent as other children, there was a risk that girls with NDDs would be overlooked. The school nurses also indicated that girls with NDDs often get their diagnoses much later than boys, which may be problematic.

### 3.3. Using an Inclusive Design for Health Education

What children with NDDs feel good about is usually good for all children. The school nurses told how they made every effort to adapt their regular teaching about, for example, ill mental health, so that it is also suitable for the students in the class who have an NDD. The school nurses often expressed that the structure and clarity that is necessary for students with NDDs also helps other students. It can be about having clear reading instructions, using image support, or combining written and oral information.
It requires some preparation and structure. The lecture must be more scheduled with planned breaks. If you adapt to the needs of the students with NDD, they can succeed as well as anyone else. (Focus group No 7)

The school nurses discussed different methods used to promote health and prevent ill health. For example, they offer lessons on nutrition, exercise, and sleep in their classes. They also talk about how to be a good friend, how to get along with each other, social networking, self-esteem and self-confidence, and stress. It is important for them to make sure that the lessons are suitable for all students in the classroom, since there are usually some children with NDDs in each class.

A way the school nurses used an inclusive design was to focus on accomplishments and things that went well, rather than focusing on problems and mistakes. The school nurses emphasized the importance of highlighting students′ strengths during the health dialogues. This applies to all children, but especially children with NDDs.

The school nurses said that another strategy to use an inclusive design was to be very explicit and clear. Students with NDDs may need more guidance when it comes to managing pain and taking medication. Children with NDDs also have more difficulty with impulse control, and the students need to learn to trust themselves and think through the different steps. Students with NDDs need a little more time to learn and to make assessments such as, “Okay, how have I slept? How have I eaten? What can I change?” before arriving to the conclusion that they need to take an aspirin for a headache. Students with NDDs would like to have a quick fix; others would too, but children with NDDs have extra difficulty with impulse control and emotion regulation, and here, the school nurse plays the important role of moderator.

The school nurses felt that using an inclusive design also involved normalizing. They thought that there was a high level of stress in society in general, and that they played an important role in helping to sort the thoughts and explain how the body works and what happens if you do not sleep or eat or move.

## 4. Discussion

The purpose of this study was to describe school nurses’ experiences of supporting students with NDDs. Three themes were identified: helping students with NDDs to interpret sensations, detecting early signs of distress among students with NDDs, and using an inclusive design for health education.

One key finding of this study was the school nurse’s role in helping students with NDDs to interpret their sensations. This is in line with the study by Bellando and Lopez [4] showing that school nurses have to start with characterizing the present problem and involve the students when doing so [4]. Caring for a child with an NDD can be demanding and present many challenges. It is important to pay attention not merely to the schoolchild in an academic context, but to problems they might face at home [17]. The school nurse plays a pivotal role in maintaining the physical and psychological well-being of students, which may alleviate the burdens placed on the student’s family [6]. School nurses strive to make the school a safe place for children who are facing difficulties [18]. Numerous school nurses in our study confirmed that when students with NDDs were under stress, they often chose to visit the school nurse, whether to discuss a particular topic, have a casual conversation, or simply for silent companionship; the main purpose was to provide a safe environment for students. The school environment can have a profound impact on several behaviors for students with NDDs, and relationships with the school’s personnel can be of particular importance [19].

Another key finding in this study was that school nurses tried to detect early signs of distress among students with NDDs. There is a need to identify and address social skills, positive emotions and positive affect, and emotional regulation for individuals who have ASD as early as possible [3]. Structured online peer mentorship programs help to improve social engagement and participation in life situations of youth with neurodevelopmental disabilities [20].

The school nurses in our study tried to use an inclusive design for health education. This is in line with the Universal Design for Learning (UDL) [21], a framework that can support the needs of all students, including those with disabilities as well as culturally and linguistically diverse learners. The focus of UDL is to build support proactively into lesson goals, curriculum resources, instructional practices, and assessments. The UDL is not a “one-size-fits-all” approach; rather, it is a framework that focuses on providing options that can meet the needs of a range of learners by building flexibility into curriculum and instruction [21]. A recent systematic review found that UDL-based instruction has the potential to increase engagement and access to students with disabilities and improve academic and social outcomes for students [22]. One area that would benefit from UDL is PE. Often students with ASD have difficulties in performing activities to meet the required PE standards. It is therefore imperative that school nurses be aware of the many challenges students with ASD bring into a PE class. School nurses provide education for the members of the school community, including the individualized education plan team, regarding the need for attention to limitations, including physical activity, of students with ASD [23].

School nurses are adept at identifying each student’s individual needs and are an indispensable resource for all students, but especially those with an NDD [4]. Students can visit the nurses for a quick break, for relieving stress, or for having a conversation with a trusted adult. The school environment can have a marked impact on several behaviors [19,24]. Improved collaboration between all involved parties is needed. These alliances involve the school and the home, as well as the relationship between the school, home, and the child and the adolescent psychiatric clinic. Even if students share a diagnosis, they should be treated individually.

## 5. Strengths and Limitations

The rich data provided by 35 school nurses in seven focus groups interviews strengthen the study. Focus group interviews have the benefit of delving deeper into a subject due to the interaction. However, the risk with focus group interviews is that participants do not dare communicate freely and speak their own minds, and therefore, further individual interviews could be a suggestion for future research. The purpose was to have one moderator and one observer in each focus group, and this was the case in five focus group interviews. Due to practical reasons, we only had one moderator but no observer in the two last sets of focus group interviews. Trustworthiness was strengthened through the analysis with five researchers. The question regarding reflexivity is always present in qualitative research. Three of the authors had a background in school nursing and had worked with children with NDDs, but two authors lacked this pre-understanding, and therefore, fruitful discussions among the authors arose during the analysis process. Only 35 out of 100 school nurses responded to the invitation to participate in the focus group interviews. Due to ethical reasons, we did not demand non-participating school nurses to give a reason for declining; however, some school nurses claimed time restraints and lack of experience as reasons for not taking part in the research. The working experience of the participating school nurses ranged from a few to more than 30 years; however, the participating school nurses may not be completely representative of the school nursing community. It could be that the school nurses that chose to participate in the study had a greater interest in children with NDDs than their colleagues. This study has the perspective of school nurses in public schools in rural and urban areas in Southern Sweden. There is a need for further studies in other regions, as well as interviewing students with NDDs and their parents/legal guardians to hear their perspective on how to provide best support to students with NDDs.

## 6. Implications for School Health Care

The result of this study implies that important tasks for school health providers are to help students with NDDs to interpret sensations, detect early signs of distress, as well as create an inclusive design for health education. UDL [21,22] is a framework that can support the need of all students, including those with NDDs. Establishing individual health and educations plans, as described by Bellando and Lopez [4], provides a structured framework for integrating this information into a comprehensive plan for students with NDDs. Time for reflection, mentorship and continuous education are also crucial for professional development [25].

## 7. Conclusions

This study highlights the importance of school nurses in identifying the needs of students with NDDs and promotes a person-centered approach to achieve a healthy and safe learning environment for all students.

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
