# Peer review of "Supporting Students with Neurodevelopment Disorders in School Health Care—School Nurses’ Experiences"

_ijerph, 2020, doi:10.3390/ijerph17165752_

Round 1
Reviewer 1 Report
In this qualitative work the authors investigate the experiences of school nurses in taking care of children with neurodevelopmental disorders. Overall, this is a very interesting article that is nicely structured and good to read, and contributes new knowledge to the debate. In the following I shortly comment on each section:
The Introduction gives a good outline of the relevance of NDDs. What I – as a reader not familiar with the Swedish school system – miss, is a short explanation about school nurses. Who are they, what is their job, are they available in every school, who pays them etc.? Some minor comments:
P1L16: Your description of the methods is accurate, still when reading superficially your wording creates the impression that you did 35 interviews in addition to 7 focus groups. Maybe you could avoid this by writing: “A qualitative method consisting of seven focus groups (that included a total of 35 school nurses) …”?
P1L40: In this sentence it seems that a word is missing after „severe”, e.g. severely impaired to…
P2L52: The subclause aims at giving examples for special needs (not for children with special needs), therefore the reference to “refugees” is linguistically incorrect. I would solve this by writing: “with special needs, such as children with mental health problems or refugees, has been found…”
For the Methods section I have some smaller comments where I would suggest adding some more details on the methodology. They are listed below:
P2L78: The description of the sampling leading to the focus groups should be more detailed. For instance, I would suggest that you also explain how many nurses you contacted initially (and where you got their contacts from), how many of those replied, and how you went about selecting your participants from those who replied. Also, it would be interesting to know how you decided about the composition of each focus group, i.e. who is going to which group.
P3L89: I recommend describing in more detail how you generated the data that was analyzed. Did you audio- or video-record the discussions? Did you transcribe them? Who did the transcription following which rules etc.?
P3L93: How did the authors derive the meaning units? Did they follow a standardized procedure (such as Krippendorff)?
The Results section is nicely structured and very informative. I have only one minor comment:
P4L166: I am not sure if I understand what the authors want to say here. The phrase “something that has been done” seems very vague to me. Could you specify what that means?
The Discussion is a bit superficial for my taste, I would recommend interconnecting your findings with the literature more closely.
The Limitations section actually does not mention any limitations. You explicitly generalize your findings to all of Sweden, and I wonder if this is right. Can you be sure that your sample is representative for all school nurses? Can you really exclude any selection bias? What do you think about the issue of reflexivity, i.e. might your study setting influence the discussions you got in the focus groups? I would like to encourage a more detailed discussion of such considerations.
P6L255: Which other contexts do you refer to? How would contexts other than school contribute to your question?
The Implications and Conclusions should be completely rewritten. See below:
P6L259-265: I have the impression that this paragraph is somewhat independent from the rest of the article, and I would contest if it is scientifically sound to present it as resulting from your data. I surely agree with what you say there, but I don’t think you can conclude it from your analysis. It starts with a methodological problem: By interviewing nurses you simply cannot answer the question if they “are adept at identifying each student’s individual needs”; that would require some gold standard of needs, and more information on how nurses identify students’ need, and a comparison of the two. Also, from your data you cannot tell if the nurses are an “indispensable resource”, because you have to rely on their own words for it. Who would not think to be an indispensable resource?
I strongly suggest you revise all of the implications section and focus on what you really found in your analysis.
P6L267-274: The same applies for the conclusion: Most of your conclusion is not a result of your research but are rather general statements concerning the usefulness of school nurses. I don’t mean to imply that their usefulness is debatable, only it is not a result of your study. As above, I suggest to completely revise the conclusion and to develop conclusions that are really based on your results.
Author Response
Response to comments from Reviewers
Manuscript ID: ijerph-878802 entitled “Supporting students with neurodevelopment disorders in school health care – school nurses’ experiences” submitted to section Health Care Sciences & Services, special issue Child and Adolescent Health Care in International Journal of Environmental Research and Public Health.
Thank you for your thoughtful comments. We have responded to each comment in italics to facilitate recognition. The changes in the manuscript are marked with yellow.
Reviewer 1
In this qualitative work the authors investigate the experiences of school nurses in taking care of children with neurodevelopmental disorders. Overall, this is a very interesting article that is nicely structured and good to read, and contributes new knowledge to the debate. In the following I shortly comment on each section: The Introduction gives a good outline of the relevance of NDDs. What I – as a reader not familiar with the Swedish school system – miss, is a short explanation about school nurses. Who are they, what is their job, are they available in every school, who pays them etc.?
Response: We have now added more information about the Swedish school system and school nursing.
Some minor comments:
P1L16: Your description of the methods is accurate, still when reading superficially your wording creates the impression that you did 35 interviews in addition to 7 focus groups. Maybe you could avoid this by writing: “A qualitative method consisting of seven focus groups (that included a total of 35 school nurses) …”?
Response: We have changed the wording according to your suggestion.
P1L40: In this sentence it seems that a word is missing after „severe”, e.g. severely impaired to…
Response: We have changed the wording according to your suggestion.
P2L52: The subclause aims at giving examples for special needs (not for children with special needs), therefore the reference to “refugees” is linguistically incorrect. I would solve this by writing: “with special needs, such as children with mental health problems or refugees, has been found…”
Response: We have changed the wording according to your suggestion.
For the Methods section I have some smaller comments where I would suggest adding some more details on the methodology. They are listed below: P2L78: The description of the sampling leading to the focus groups should be more detailed. For instance, I would suggest that you also explain how many nurses you contacted initially (and where you got their contacts from), how many of those replied, and how you went about selecting your participants from those who replied. Also, it would be interesting to know how you decided about the composition of each focus group, i.e. who is going to which group.
Response: We have now modified the methods section according to your suggestions. The sampling is now illustrated (100 school nurses were invited, of which 35 choose to participate in the study), as well as the composition of the focus groups (the focus groups consisted of school nurses from the same municipality and they therefore knew each other).
P3L89: I recommend describing in more detail how you generated the data that was analyzed. Did you audio- or video-record the discussions? Did you transcribe them? Who did the transcription following which rules etc.?
Response: We have now modified the methods section according to your suggestions. The interviews were audio recorded, and transcribed verbatim by the first two authors.
P3L93: How did the authors derive the meaning units? Did they follow a standardized procedure (such as Krippendorff)?
Response: We have now modified the methods section according to your suggestions. We followed the standardized procedure described by Krippendorff.
The Results section is nicely structured and very informative. I have only one minor comment: P4L166: I am not sure if I understand what the authors want to say here. The phrase “something that has been done” seems very vague to me. Could you specify what that means?
Response: This sentence has been deleted.
The Discussion is a bit superficial for my taste, I would recommend interconnecting your findings with the literature more closely.
Response: We have now reworked the discussion section with more interconnection between our findings and the literature.
The Limitations section actually does not mention any limitations. You explicitly generalize your findings to all of Sweden, and I wonder if this is right. Can you be sure that your sample is representative for all school nurses? Can you really exclude any selection bias? What do you think about the issue of reflexivity, i.e. might your study setting influence the discussions you got in the focus groups? I would like to encourage a more detailed discussion of such considerations.
Response: We have now modified the limitation section according to your suggestions.
P6L255: Which other contexts do you refer to? How would contexts other than school contribute to your question?
Response: We have now reworded this sentence.
The Implications and Conclusions should be completely rewritten. See below: P6L259-265: I have the impression that this paragraph is somewhat independent from the rest of the article, and I would contest if it is scientifically sound to present it as resulting from your data. I surely agree with what you say there, but I don’t think you can conclude it from your analysis. It starts with a methodological problem: By interviewing nurses you simply cannot answer the question if they “are adept at identifying each student’s individual needs”; that would require some gold standard of needs, and more information on how nurses identify students’ need, and a comparison of the two. Also, from your data you cannot tell if the nurses are an “indispensable resource”, because you have to rely on their own words for it. Who would not think to be an indispensable resource?
I strongly suggest you revise all of the implications section and focus on what you really found in your analysis.
Response: We have now rewritten the Implications and Conclusions completely according to your suggestions.
P6L267-274: The same applies for the conclusion: Most of your conclusion is not a result of your research but are rather general statements concerning the usefulness of school nurses. I don’t mean to imply that their usefulness is debatable, only it is not a result of your study. As above, I suggest to completely revise the conclusion and to develop conclusions that are really based on your results.
Response: We have now rewritten the Implications and Conclusions completely according to your suggestions.
Reviewer 2 Report
It is considered that the study deals with current and relevant thematic for the area of knowledge in nursing, education and school health.
The introduction is generally well-founded, contextualizing the theme and presenting a logical sequence. However, a more in-depth and critical review would be important, based on the current state of the art of the object of study, that is, the work and performance, specifically of school health nurses, in the care of students with neurological disorders, in order to provide a better basis and problematize it. How has this work been implemented? What scientific literature and protocols refer to the support to be given by nurses to students with NDD?
The method is appropriate to the type of study and presents adequate information, but it is suggested to better describe the operationalization and realization of the focus groups, especially about the adequacy of spaces for the groups and their conduct. It was said that they were carried out in the workplace, but how were the schedules agreed? Did it interfere with the nurses' work? Were adequate, private rooms? The ideal is a room that comfortably accommodates the expected number of participants and moderators and that is protected from noise and external interruptions. Were the interviews recorded? Or were the interviews recorded by the mediator? How did this record take place? Were they transcribed, in the same way, as, by whom? Were electronic equipment required, use of recorders, for example, what type? What determined the realization of seven focus groups? Was it to explore the theme under study or an operational issue?
The authors do not provide data regarding the moderator and the dynamics of the groups, two aspects that are especially important for measuring the appropriate use of the focus group technique. Who led the groups? With regard to the moderator, a starting condition is that he has substantial knowledge of the topic under discussion so that he can conduct the group properly. In addition to the moderator, was there support from a second moderator?
Author Response
Response to comments from Reviewers
Manuscript ID: ijerph-878802 entitled “Supporting students with neurodevelopment disorders in school health care – school nurses’ experiences” submitted to section Health Care Sciences & Services, special issue Child and Adolescent Health Care in International Journal of Environmental Research and Public Health.
Thank you for your thoughtful comments. We have responded to each comment in italics to facilitate recognition. The changes in the manuscript are marked with yellow.
Reviewer 2
It is considered that the study deals with current and relevant thematic for the area of knowledge in nursing, education and school health. The introduction is generally well-founded, contextualizing the theme and presenting a logical sequence. However, a more in-depth and critical review would be important, based on the current state of the art of the object of study, that is, the work and performance, specifically of school health nurses, in the care of students with neurological disorders, in order to provide a better basis and problematize it. How has this work been implemented? What scientific literature and protocols refer to the support to be given by nurses to students with NDD?
Response: We have now modified the introduction according to your suggestions.
The method is appropriate to the type of study and presents adequate information, but it is suggested to better describe the operationalization and realization of the focus groups, especially about the adequacy of spaces for the groups and their conduct. It was said that they were carried out in the workplace, but how were the schedules agreed? Did it interfere with the nurses' work? Were adequate, private rooms? The ideal is a room that comfortably accommodates the expected number of participants and moderators and that is protected from noise and external interruptions. Were the interviews recorded? Or were the interviews recorded by the mediator? How did this record take place? Were they transcribed, in the same way, as, by whom? Were electronic equipment required, use of recorders, for example, what type? What determined the realization of seven focus groups? Was it to explore the theme under study or an operational issue?
Response: We have now modified the methods according to your suggestions. The interviews took place in adequate rooms protected from noise and external interruptions. The interviews were audio recorded and transcribed verbatim by the first two authors. In the last two sets of interviews, no new data was obtained, and therefore 7 focus groups were considered sufficient.
The authors do not provide data regarding the moderator and the dynamics of the groups, two aspects that are especially important for measuring the appropriate use of the focus group technique. Who led the groups? With regard to the moderator, a starting condition is that he has substantial knowledge of the topic under discussion so that he can conduct the group properly. In addition to the moderator, was there support from a second moderator?
Response: We have now modified the methods according to your suggestions. Four of the authors took turns in being moderator and/or observer in the focus group interviews. All authors were registered nurses with extensive knowledge about NDD, and three of the authors had a background in research and school nursing.
Reviewer 3 Report
Dear author,
The paper does not raise central concerns. However, I would hope to hear from about some questions:
- Participants recruitment: what are selection criteria/ strategies to define the sample? Please state it clear.
- Line 89, Graneheim and Lundman (2004) seems to be missing in References list.
- Discussion: it would be clearer if you deeply discuss results, based on literature, onto each of the three dimensions/ themes identified in results section. Do you think the paper reach your goals and intentions (lines 55-58)?
Best regards.
Author Response
Response to comments from Reviewers
Manuscript ID: ijerph-878802 entitled “Supporting students with neurodevelopment disorders in school health care – school nurses’ experiences” submitted to section Health Care Sciences & Services, special issue Child and Adolescent Health Care in International Journal of Environmental Research and Public Health.
Thank you for your thoughtful comments. We have responded to each comment in italics to facilitate recognition. The changes in the manuscript are marked with yellow.
Reviewer 3
The paper does not raise central concerns. However, I would hope to hear from about some questions:
Participants recruitment: what are selection criteria/ strategies to define the sample? Please state it clear.
Response: We have now modified the methods according to your suggestions. The sampling procedure is now better described.
Line 89, Graneheim and Lundman (2004) seems to be missing in References list.
Response: We have now added the reference in the list.
Discussion: it would be clearer if you deeply discuss results, based on literature, onto each of the three dimensions/ themes identified in results section. Do you think the paper reach your goals and intentions (lines 55-58)?
Response: We have now modified the discussion section according to your suggestions.